The efficacy of commercial decontamination agents differs between standardised test settings and research laboratory usage for a variety of bacterial species

Uy Benedict 1
Read Hannah 2
van de Pas Shara 2
Marnane Rebecca 1
Casu Francesca 3
http://orcid.org/0000-0001-7352-1112 Swift Simon 4
Wiles Siouxsie 2 5 s.wiles@auckland.ac.nz
1 Technical Services, University of Auckland , Auckland , New Zealand
2 Bioluminescent Superbugs Lab, Department of Molecular Medicine and Pathology, University of Auckland , Auckland , New Zealand
3 Health Safety and Wellbeing, University of Auckland , Auckland , New Zealand
4 Department of Molecular Medicine and Pathology, University of Auckland , Auckland , New Zealand
5 Te Pūnaha Matatini , Auckland , New Zealand
Silva Pedro
Electronic publication date: 2022 Jul 15
Publication date: 2022
Volume: 10
Electronic Location ID: e13646
Received 2022 Feb 23; Accepted 2022 Jun 7
Copyright: © 2022 Uy et al.
Copyright year: 2022
Copyright holder: Uy et al.
License: This is an open access article distributed under the terms of the Creative Commons Attribution License, which permits unrestricted use, distribution, reproduction and adaptation in any medium and for any purpose provided that it is properly attributed. For attribution, the original author(s), title, publication source (PeerJ) and either DOI or URL of the article must be cited.
License URL: https://creativecommons.org/licenses/by/4.0/

Keywords: Escherichia coli, Staphylococcus aureus, Mycobacterium tuberculosis, Disinfection, Mycobacteria, Bleach, Trigene, Pseudomonas aeruginosa, Disinfecting agents, Bacillus

Funding: School of Medical Sciences, University of Auckland This work was supported by internal funds from the School of Medical Sciences at the University of Auckland. The funders had no role in study design, data collection and analysis, decision to publish, or preparation of the manuscript.

==============================
Decontamination of surfaces and items plays an important role in reducing the spread of infectious microorganisms in many settings including hospitals and research institutes. Regardless of the location, appropriate decontamination procedures are required for maintaining biosafety and biosecurity. For example, effective decontamination of microbial cultures is essential to ensure proper biocontainment and safety within microbiological laboratories. To this end, many commercial decontamination agents are available which have been tested to a prescribed standard to substantiate their efficacy. However, these standardised tests are unlikely to accurately reflect many conditions encountered in medical and biomedical research. Despite this, laboratory workers and other users of decontamination agents may assume that all decontamination agents will work in all situations. We tested commonly used commercial decontamination agents against a range of bacterial species to determine their efficacy under real-world research laboratory conditions. As each decontamination agent has a different recommended dilution for use, to compare their efficacy we calculated their ‘effective ratio’ which reflects the difference between the manufacturer-recommended dilution and the dilution needed to achieve decontamination under real-world research laboratory conditions. Effective ratios above one indicate that the agent was effective at a dilution more dilute than recommended whereas effective ratios lower than one indicate that the agent required a higher concentration than recommended. Our results show that the quaternary ammonium agents TriGene Advance and Chemgene HLD4L were the most effective out of the agents tested, with biocidal activity measured at up to 64 times the recommended dilution. In contrast, hypochlorite (bleach) and Prevail™ (stabilised hydrogen peroxide) had the lowest effective ratios amongst the tested agents. In conclusion, our data suggests that not all decontamination agents will work at the recommended dilutions under real-world research laboratory conditions. We recommend that the protocols for the use of decontamination agents are verified under the specific conditions required to ensure they are fit for purpose.

Introduction

Decontamination of surfaces and items plays an important role in reducing the spread of infectious microorganisms in many settings including hospitals and research institutes. Within microbiological laboratories, the effective decontamination of microbial cultures is essential to ensure that users are not exposed to the microorganisms being manipulated within the facility and that the microorganisms are not inadvertently released into the environment. This applies to laboratories involved in biomedical research as well as diagnostic laboratories and other testing facilities. Although each type of facility will have different samples and decontamination applications, all will have guidelines to prevent and reduce the occurrence of containment breaches. Proper decontamination procedures are a part of these guidelines.

Within different research facilities, there are institution-specific decontamination procedures for various types of contaminated materials which include both autoclaving and chemical decontamination. Six main classes of chemical decontamination agents are routinely used: iodophors, quaternary ammonium compounds (QAC), peroxides, phenols, chlorine, and aldehydes (Chapman, 2003). Each class has a different mechanism of action and their own set of advantages and disadvantages (Geraghty et al., 2014). The type of agent that is used can also depend on the application; for example, aldehydes are generally used for the decontamination of medical devices, while iodophors are generally used as antiseptics (Bell et al., 1989).

Decontamination agents of the different classes are commercially available, and companies selling these agents verify the bactericidal claims through independent standardised testing by certified laboratories. The specific standards for these tests include a pre-determined set of conditions used to test their efficacy against specific microorganisms. These standards can be for a continent such as the European Standards (EN) or a specific country such as the American Society for Testing and Material (ATSM), and the Environmental Protection Agency (EPA) standards for the United States, or the Canadian standards from the Standards Council of Canada. There are also international standards such as those from the Association of Official Analytical Chemists (AOAC International) and the Clinical & Laboratory Standards Institute (CLSI). For each standard, there are separate bactericidal, sporicidal, mycobactericidal, and fungicidal tests. For example, European Standard EN 1276 is the standard for evaluating the bactericidal activity of chemical disinfectants and antiseptics used in food, industrial, domestic and institutional areas, whereas EN 1650 is the fungal equivalent. The standards also differ in their purpose; EN 1656 is tailored for determining bactericidal activities in veterinary areas, while EN 13623 investigates bactericidal activity in aqueous systems. Similarly, EN 1276 is appropriate for testing materials in suspension whereas EN 13697 is appropriate for non-porous surfaces.

While the name of each standard offers an indication of the testing aims, the detailed protocols are often hidden behind a paywall making it difficult to ascertain how each test is performed. Despite this, decontamination agent users may assume that the conditions used for each standard by each company would be the same and thus comparable. However, this is not the case. For example, according to their websites, the companies that manufacture TriGene Advance and Chemgene HLD4L report that testing activity against Mycobacterium terrae was performed using different dilutions, holding times, and endpoints despite both using the EN 14348 standard. This is concerning as hold time and dilution are known to be important in determining the efficacy of decontamination (Rutala & Weber, 1999; Russell & McDonnell, 2000). Although not stated, it could be assumed that the inocula for both tests were similar as microbial load is known to influence decontamination efficacy (Rutala & Weber, 1999; Rutala & Weber, 2015). Also concerning is that the Chemgene product label currently recommends a lower dilution and shorter hold time for decontaminating mycobacterial-contaminated items compared to those they tested for determining efficacy.

One other important consideration is whether the endpoints used within the standards to determine antimicrobial activity are relevant for microbiological laboratories. For example, the EN 1276 bactericidal tests use a >5-log reduction to determine activity. However, when working with pure cultures of bacteria in a research laboratory, concentrations of 109 colony forming units (CFU) per mL are often reached with total volumes ranging from less than 5 mL to several litres. Under optimum conditions, many fast-growing bacterial species can reach these concentrations within 24 h. This means that to be effective in practice, decontamination agents must be able to kill a larger number of cells and achieve more than the 5-log reduction set in the EN 1276 standard. Other considerations include whether the media the bacteria are grown in contain proteins that will interfere with the efficacy of the decontamination agents which in the standards are tested using specific defined media.

Within the research laboratories at the University of Auckland, most solid wastes such as agar plates and contaminated plasticware are autoclaved while liquid waste and small contaminated items such as pipette tips are chemically decontaminated. In this study, we investigated the efficacy of commonly-used and commercially-available decontamination agents against a variety of pathogens, including Gram-positive, Gram-negative, and Mycobacterial spp. These agents were serially diluted below their recommended dilution to determine how well they retain their antibacterial efficacy through controlled neutralisation. Importantly, our tests mimic the decontamination of contaminated liquid laboratory wastes generated in standard microbiological research laboratories, using higher bacterial concentrations and more complex media than used in the standards designed to verify efficacy. In addition, we investigated the efficacy of these decontamination agents against antibiotic-resistant strains of Escherichia coli which we hypothesised could be more resistant to killing by the decontamination agents due to cross resistance (Tattawasart et al., 1999; Wesgate, Grasha & Maillard, 2016). Our results show that not all decontamination agents will work at the recommended dilutions under real-world laboratory conditions.

Materials and Methods

Bacterial strains

The bacterial strains used in this study are shown in Table 1 and were purchased from ATCC through In Vitro Technologies. Bacteria were revived from frozen stocks stored at −80 °C prior to use to prevent adaptation over multiple laboratory subcultures. Bacteria were grown to stationary phase using the appropriate liquid media and temperature (Table 1). All growth media and supplements were purchased from Fort Richard (New Zealand). Liquid cultures of mycobacterial species were grown in Middlebrook 7H9 broth supplemented with 10% Middlebrook ADC enrichment media, 0.4% glycerol (Sigma-Aldrich, Auckland, New Zealand), and 0.05% Tween 20 (Sigma-Aldrich, Auckland, New Zealand). Mycobacterium smegmatis and M. abscessus were grown at 37 °C for 42–48 h while M. marinum was grown at 28 °C for 10–14 days. M. tuberculosis was grown for 4–6 weeks at 37 °C in a Biosafety Level 3 (BSL3) laboratory. All other cultures were grown for 18–24 h. Stationary phase Mycobacterial cultures were adjusted to an optical density at 600 nm (OD600) of 1 in 7H9 broth supplemented with 10% ADC enrichment media. All other bacteria were adjusted to an OD600 of 1 in BD BBL™ cation-adjusted Mueller-Hinton II Broth (MHB). Adjusted cultures were serially diluted and plated on solid media to determine the initial inocula. For Mycobacterial cultures Middlebrook 7H10 agar supplemented with 10% Middlebrook OADC enrichment media was used. All other bacteria were grown on Mueller-Hinton II Agar. Concentrations ranged from ~108–109 CFU/mL, depending on the organism. For M. marinum, M. smegmatis, and M. tuberculosis the viable counts returned were 10–100-fold lower, but this reflects the tendency of this method to underestimate the number of viable bacteria present due to the propensity of these organisms to aggregate.

Table 1 Bacterial strains used in this study.

Species	Strain	Description	Growth medium	Temperature (°C)	Ref.	
Acinetobacter baumannii	BAA-1790	Multidrug-resistant; isolated from sputum	Tryptic soy	37	Korotetskiy et al. (2020)	
Bacillus spizizenii	ATCC 6633	Formerly B. subtilis; quality control strain	Brain heart infusion	30	ATCC	
Escherichia coli	ATCC 25922	Quality control strain	Tryptic soy	37	Minogue et al. (2014)	
BAA-2340	Carbapenem-resistant KPC reference strain	Nutrient	37	Trepanier et al. (2017)	
BAA-2452	Carbapenem-resistant NDM-1 reference strain	Tryptic soy	37	Trepanier et al. (2017)	
Top 10	Antibiotic-susceptible molecular cloning strain	Luria-Bertani	37	ThermoFisher Scientific	
Klebsiella oxytoca	ATCC 49131	Quality control strain	Tryptic Soy	37	ATCC	
Mycobacterium abscessus	NZRM 4048	Clinical isolate	7H9	37	Freeman et al. (2007)	
Mycobacterium marinum	ATCC 927	Type strain; isolate from fish	7H9	30	Aronson (1926)	
Mycobacterium smegmatis	ATCC 700084	Sensitive to kanamycin; mc2155	7H9	37	Snapper et al. (1990)	
Mycobacterium tuberculosis	ATCC 25618	Virulent; H37Rv	7H9	37	Oatway & Steenken (1936)	
Pseudomonas aeruginosa	ATCC 27853	Quality control strain	Tryptic soy	37	Cao et al. (2017)	
Staphylococcus aureus	ATCC 6538	Quality control strain	Tryptic soy	37	Makarova et al. (2017)	

Decontamination agent preparation

Commercial decontamination agents were purchased from their respective suppliers (Table 2) and were tested at a wide range of concentrations which included their recommended dilution. The agents were diluted in 7H9 broth supplemented with 10% ADC for the mycobacterial strains, and in MHB for all other bacteria.

Table 2 Decontamination agents used in this study.

Decontamination agent	Manufacturer	Supplier	Active component	Recommended dilution	
Bleach1	Brighton professional	Staples New Zealand Ltd, Auckland, New Zealand	Sodium hypochlorite	1:10	
HLD4L	Chemgene	ThermoFisher Scientific, Auckland, New Zealand	Surfactant with quaternary ammonium compounds	1:100	
Prevail™	Virox	Anissentials, Auckland, New Zealand	Hydrogen peroxide	1:40	
TriGene Advance	Trisiel	In Vitro Technologies, Auckland, New Zealand	Poly biguanide with quaternary ammonium compounds	1:100	
Virkon	Dupont	ThermoFisher Scientific, Auckland, New Zealand	Oxone, benzenesulfonate and sulfamic acid mixture	1:100 (as a 1% w/v solution)	
Note:

1 4% solution of stabilised liquid bleach with 53 g/L of hypochlorite.

Measurement of bactericidal activity of decontaminating agents

The bactericidal activity of each decontaminating agent was measured using a variation of the microdilution method in a microtiter plate (Wiegand, Hilpert & Hancock, 2008) using two technical replicates for each of 3–5 biological replicates. We added 200 µL of decontamination agent in duplicate to the top wells of a clear, flat-bottomed 96-well plate (Nunc; Thermo Scientific, Auckland, New Zealand) and filled the remaining wells with 100 µL of either 7H9 for mycobacteria or MHB for all other species. 100 µL of each decontamination agent was then serially diluted down the 96-well plate to give doubling dilutions. If greater dilutions were needed, the agent was initially diluted in the appropriate media (Table 1) before adding to the top wells of the microtiter plate. Then we added 100 µL of diluted bacterial culture to each well and incubated the plates static at room temperature. At various time points over 4 h, 4 × 10 µL aliquots were removed from each well and plated onto the appropriate agar plates. These were incubated at the appropriate temperature/time combination depending on the bacterial strain being tested and any growth determined by eye. The Minimum Bactericidal Dilution Factor (MBDF) was calculated as the greatest dilution of disinfectant returning no visible bacterial growth. As previously indicated, experiments were repeated 3–5 times.

Effective ratio calculation

Due to the different recommended dilutions for the decontamination agents, an Effective Ratio was calculated by dividing the MBDF by the recommended dilution (Table 2). An effective ratio of one indicates the agent is effective at the recommended dilution. A ratio greater than one indicates that the decontamination agent is effective at a dilution more dilute than recommended. An effective ratio lower than one indicates that the agent is not effective at the recommended dilution.

Area under curve calculation

AUC values (as Minimum Bactericidal Dilution Factor x holding time (min)) were calculated for each agent-organism combination from the Minimum Bactericidal Dilution Factor values for each of the five holding times using GraphPad Prism version 8.4.3.

Statistical analysis

Statistical analysis was performed using GraphPad Prism version 8.4.3. Data was analysed using a mixed-effects model with Tukey’s multiple comparison test. Statistical significance was set at p ≤ 0.05.

Results

We investigated the bactericidal activity of bleach, Chemgene HLD4L, Prevail™, TriGene Advance, and Virkon against a variety of Gram-positive, Gram-negative, and Mycobacterial species. The minimum dilutions we observed as necessary for bactericidal activity (given as median and range) are presented in Table 3. Due to the decontamination agents having different recommended dilutions for use (Table 2) (Gélinas & Goulet, 1983), to easily visualise the efficacy of each agent we calculated an Effective Ratio for each bacterium-decontamination agent combination (Fig. 1). This was achieved by dividing the minimum bactericidal dilution by the recommended dilution. An Effective Ratio of one indicates an agent is effective at the recommended dilution. A ratio greater than one indicates that an agent is effective at a dilution more dilute than recommended while a ratio lower than one indicates that an agent is not effective at the recommended dilution.

Table 3 Minimum bactericidal dilutions for various decontaminating agents at different holding times.

Species	Strain	Holding time (min)	Minimum bactericidal dilution (median and range)	
Bleach	HLD4L	Prevail™	TriGene	Virkon	
Acinetobacter baumannii	BAA-1790	10	1:8 (1:8–1:16)1	1:1,600	1:40	1:3,200 (1:1,600–1:3,200)	1:400 (1:200–1:400)	
30	1:16 (1:16–1:32)	1:1,600 (1:1,600–1:3,200)	1:40	1:3,200 (1:3,200–1:6,400)	1:400 (1:200–1:400)	
60	1:32 (1:32–1:64)	1:1,600 (1:1,600–1:3,200)	1:40	1:3,200 (1:3,200–1:6,400)	1:400 (1:200–1:400)	
240	1:64	1:3,200 (1:1,600–1:3,200)	1:40	1:3,200 (1:3,200–1:6,400)	1:400	
1,440	1:64	1:1,600 (1:1,600–1:3,200)	1:80 (1:40–1:80)	1:6,400	1:400	
Bacillus spizizenii	ATCC 6633	10	1:2	1:3,200	1:640 (1:256–1:640	1:3,200 (1:1,600–1:3,200)	1:40	
30	1:2	1:3,200	1:320 (1:256–1:640)	1:3,200 (1:1,600–1:3,200)	1:40	
60	1:2	1:3,200	1:640 (1:256–1:640)	1:3,200 (1:1,600–1:3,200)	1:40 (1:40–1:80)	
240	1:2	1:3,200	1:640 (1:256–1:640)	1:3,200 (1:1,600–1:6,400)	1:80	
1,440	1:2	1:3,200	1:640 (1:256–1:1,280)	1:3,200 (1:1,600–1:3,200)	1:80 (1:80–1:160)	
Escherichia coli	ATCC 25922	10	1:10 (1:10–1:16)	1:400 (1:400–1:800)	1:40 (1:40–1:80)	1:1,600 (1:1,600–1:3,200)	1:200 (1:100–1:200)	
30	1:10 (1:8–1:20)	1:800	1:80	1:3,200 (1:1,600–1:3,200)	1:200 (1:200–1:400)	
60	1:32 (1:20–1:40)	1:800	1:80	1:3,200	1:200 (1:200–1:400)	
240	1:40 (1:40–1:64)	1:1,600 (1:800–1:1,600)	1:80	1:3,200 (1:3,200–1:6,400)	1:200 (1:200–1:400)	
1,440	1:40 (1:40–1:64)	1:3,200	1:80	1:6,400	1:200 (1:200–1:400)	
	BAA-2340	10	1:8 (1:8–1:32)	1:400	1:80 (1:40–1:80)	1:1,600	1:200	
30	1:32	1:400	1:80 (1:40–1:80)	1:1,600	1:200	
60	1:32 (1:32–1:64)	1:400 (1:400–1:800)	1:80	1:3,200 (1:1,600–1:3,200)	1:200 (1:200–1:400)	
240	1:64 (1:32-1:64)	1:800	1:80 (1:80–1:160)	1:3,200	1:400	
1,440	1:64 (1:32–1:64)	1:1,600 (1:800–1:1,600)	1:160 (1:80–1:320)	1:3,200 (1:3,200–1:6,400)	1:400	
	BAA-2452	10	1:10 (1:10-1:20)	1:400	1:20 (1:20–1:40)	1:800 (1:800–1:1,600)	1:80 (1:80–1:100)	
30	1:20 (1:10–1:40)	1:800	1:20 (1:20–1:40)	1:1,600	1:160 (1:160–1:200)	
60	1:20 (1:20–1:40)	1:800	1:40 (1:20–1:40)	1:1,600	1:160 (1:160–1:200)	
240	1:60 (1:20–1:80)	1:1,600	1:40 (1:40–1:80)	1:3,200	1:160 (1:160–1:200)	
1,440	1:60 (1:20–1:80)	1:2,400 (1:1,600–1:3,200)	1:40 (1:40–1:80)	1:3,200	1:160 (1:160-1:200)	
	Top 10	10	1:4 (1:4–1:8)	1:200	1:80 (1:80–1:128)	1:1,600 (1:800–1:1,600)	1:320 (1:160–1:320)	
30	1:8 (1:4–1:8)	1:400 (1:200-1:400)	1:80 (1:80–1:128)	1:1,600 (1:1,800–1:1,600)	1:320 (1:160–1:320)	
60	1:16 (1:4–1:32)	1:400	1:160 (1:128–1:160)	1:1,600 (1:1,600–1:3,200)	1:320 (1:160–1:320)	
240	1:64 (1:32–1:64)	1:800 (1:400–1:800)	1:160 (1:128–1:160)	1:3,200	1:640 (1:320–1:640)	
1,440	1:64 (1:32–1:64)	1:1600	1:160 (1:128–1:160)	1:6,400 (1:3,200–1:6,400)	1:640 (1:320–1:640)	
Klebsiella oxytoca	ATCC 49131	10	1:16 (1:16–1:32)	1:200 (1:200–1:400)	1:40	1:800	1:200 (1:100–1:200)	
30	1:32	1:400 (1:400–1:800)	1:40 (1:40–1:80)	1:1,600 (1:800–1:1,600)	1:200	
60	1:32 (1:32–1:64)	1:800	1:80	1:1,600	1:200 (1:200–1:400)	
240	1:64	1:800	1:80	1:3,200	1:400 (1:200–1:400)	
1,440	1:64	1:1,600	1:80	1:3,200	1:400 (1:200–1:400)	
Mycobacterium abscessus	NZRM 4048	10	1:8 (1:4–1:8)	1:8 (1:8–1:16)	1:8 (1:8–1:16)	1:32	NA 2	
30	1:8	1:16	1:16	1:32	1:20	
60	1:8 (1:8–1:16)	1:16	1:16	1:32	1:20	
240	1:8 (1:8–1:16)	1:16 (1:16–1:32)	1:16	1:32	1:40 (1:20–1:80)	
1,440	1:16	1:32 (1:16–1:128)	1:32	1:32 (1:32–1:64)	1:80 (1:80–1:320)	
Mycobacterium marinum	ATCC 927	10	1:8 (1:4–1:16)	1:40 (1:32–1:64)	1:16	1:104 (1:64–1:320)	NA	
30	1:12 (1:4-1:16)	1:52 (1:32–1:80)	1:16	1:104 (1:64–1:640)	NA	
60	1:12 (1:8-1:16)	1:72 (1:32–1:80)	1:16	1:144 (1:128–1:640)	NA	
240	1:16 (1:8–1:16)	1:104 (1:64–1:160)	1:24 (1:16–1:32)	1:208 (1:128–1:210)	NA	
1,440	1:16 (1:16-1:32)	1:288 (1:256–1:640)	1:32	1:288 (1:256–1:640)	1:80 (1:80–1:160)	
Mycobacterium smegmatis	ATCC 700084	10	1:16 (1:4-1:32)	1:400	1:16	1:800 (1:800–1:1,600)	1:40 (1:20–1:160)	
30	1:16 (1:8–132)	1:400 (1:400–1:800)	1:24 (1:16–1:32)	1:800 (1:800–1:1,600)	1:40 (1:20–1:320)	
60	1:16 (1:16–1:32)	1:800 (1:400–1:800)	1:32 (1:32–1:64)	1:1600 (1:800–1:1,600)	1:40 (1:20–1:320)	
240	1:32 (1:32–1:64)	1:800	1:128 (1:64–1:160)	1:1,600 (1:1,600–1:3,200)	1:80 (1:40–1:640)	
1,440	1:32 (1:32–1:64)	1:1,600	1:128 (1:128–1:160)	1:4,800 (1:3,200–1:6,400)	1:160 (1:80–1:640)	
Mycobacterium tuberculosis	ATCC 25618	10	1:2 (1:2–1:4)	1:128 (1:64–1:128)	1:16	1:128	NA	
30	1:2 (1:2–1:4)	1:64 (1:64–1:128)	1:16	1:128	NA	
60	1:2 (1:2–1:4)	1:64 (1:64–1:128)	1:16	1:128	NA	
240	1:4 (1:2-1:4)	1:64 (1:64–1:128)	1:16 (1:16–1:32)	1:128	1:40	
1,440	1:4 (1:4–1:8)	1:128	1:32	1:128	1:80	
Pseudomonas aeruginosa	ATCC 27853	10	1:16 (1:8–1:16)	1:400	1:40	1:400	1:200	
30	1:16	1:400 (1:400–1:800)	1:40	1:400 (1:400–1:800)	1:200	
60	1:32 (1:16–1:32)	1:400 (1:400–1:800)	1:40	1:800	1:400 (1:200–1:400)	
240	1:32	1:400 (1:400–1:800)	1:40	1:800	1:400	
1,440	1:32	1:800 (1:400–1:800)	1:40 (1:40–1:80)	1:800 (1:800–1:1,600)	1:400	
Staphylococcus aureus	ATCC 6538	10	1:8 (1:8–1:16)	1:1,600	1:160 (1:40–1:160)	1:3,200 (1:1,600–1:3,200)	1:400 (1:400–1:800)	
30	1:16 (1:16–1:32)	1:1,600 (1:1,600–1:3,200)	1:80 (1:40–1:160)	1:3,200	1:400 (1:400–1:800)	
60	1:16 (1:10–1:32)	1:3,200	1:160 (1:80–1:320)	1:3,200 (1:3,200–1:6,400)	1:800 (1:400–1:1,600)	
240	1:32 (1:20–1:64)	1:3,200	1:160 (1:80–1:640)	1:6,400 (1:3,200–1:6,400)	1:800 (1:400–1:1,600)	
1,440	1:64 (1:20–1:128)	1:6,400 (1:3,200–1:6,400)	1:320 (1:80–1:1,280)	1:6,400 (1:6,400–1:12,800)	1:1,600 (1:400–1:3,200)	
Notes:

1 Values shown in bold indicate a condition under which the recommended dilution would not be sufficient for decontamination.

2 Not applicable.

Figure 1 Effectiveness of various decontamination agents against a range of bacterial species at different hold times.

As each decontamination agent has a different recommended dilution for use, to compare their efficacy we calculated their effective ratios by dividing the Minimum Bactericidal Dilution Factor by the recommended dilution for each agent. An Effective Ratio (ER) of one (grey boxes) indicates an agent active at the recommended dilution, while an ER above one (dark blue boxes) indicates an agent was active at a dilution more dilute than recommended and an ER below one (orange boxes) indicates an agent that was not active at the recommended dilution. Data (n = 3–5) is presented as the median ER for each species at a given hold time. The raw data is available online from https://doi.org/10.17608/k6.auckland.19142606.

Activity of decontaminating agents against Gram-negative bacteria

We used A. baumannii, E. coli, K. oxytoca, and P. aeruginosa as representative Gram-negative organisms. As can be seen from the data provided in Table 3 and Fig. 1, Chemgene HLD4L, TriGene Advance, and Virkon were bactericidal against A. baumannii, K. oxytoca, and P. aeruginosa with Effective Ratios above one showing that these agents were effective at dilutions greater than those recommended at all holding times (Fig. 1). With Effective Ratios of one, Prevail™ was effective against these species at the recommended dilution at holding times as short as 10 min (Fig. 1). While bleach was effective against K. oxytoca at the recommended dilution for all holding times tested, in some experiments with A. baumannii and P. aeruginosa, Effective Ratios were less than one indicating that a lower dilution than that recommended was needed to achieve bactericidal activity using a holding time of 10 min (Fig. 1).

Given the increased prevalence of antibiotic-resistant bacteria, we investigated whether the decontamination agents would have similar activity against E. coli strains with different resistance profiles. ATCC 25922 is an antibiotic-sensitive strain recommended for quality control purposes while Top 10 is a commonly used molecular cloning strain that is also sensitive to antibiotics. In contrast, BAA-2340 and BAA-2452 are both carbapenem resistant. According to the ATCC website, BAA-2340 encodes a Klebsiella pneumonia carbapenemase (KPC) while BAA-2452 encodes a New Delhi Metalloprotease 1 (NDM-1). Overall, the decontamination agents were effective against the four E. coli strains with some minor differences. Bleach was effective at the recommended dilution except when used against BAA-2340 with a 10 min holding time and Top 10 with holding times of 10, 30, and 60 min (Table 3, Fig. 1) though only the 30 min hold time for Top 10 reached statistical significance (p = 0.005, two-way ANOVA with Dunnett’s multiple comparison test). Prevail™ also required a lower dilution than recommended when used against BAA-2452 with hold times of 10, 30, and 60 min (p = 0.03, Two Way ANOVA with Dunnett’s multiple comparison test).

To further determine if the overall responses of the different E. coli strains were statistically significant, we also calculated Area Under Curve values from the Minimum Bactericidal Dilution Factors obtained for each strain for each decontamination agent over the five holding times (Fig. 2). Lower AUC values reflect lower dilutions required for activity and hence increased resistance of the bacterium to the decontamination agent. We analysed this data using a mixed-effects model with Tukey’s multiple comparison test. The results indicate that Top 10 is significantly more resistant to HLD4L than ATCC 25922 (p = 0.0188), BAA-2452 is significantly more resistant to Trigene than ATCC 25922 (p = 0.0346), and that BAA-2452 is significantly more resistant to Virkon than BAA-2430 (p < 0.0001).

Figure 2 The efficacy of decontamination agents can vary between E. coli isolates.

To compare the efficacy of the decontamination agents between different E. coli isolates (ATCC 25922 (antibiotic-sensitive); BAA-2340 (carbapenem-resistant, KPC+); BAA-2452 (carbapenem-resistant, NDM-1+); Top 10 (antibiotic-sensitive, molecular cloning strain)) we calculated Area Under Curve (AUC) values from the minimum bactericidal dilutions obtained over the five holding times. Lower AUC values reflect lower dilutions required for activity and hence increased resistance of the bacterium to the decontamination agent. Data (n = 3–5) is presented as box and whisker plots with median. The data was analysed using a mixed-effects model with Tukey’s multiple comparison test. The raw data is available online from https://doi.org/10.17608/k6.auckland.19142606.

Activity of decontaminating agents against Gram-positive bacteria

We used B. spizizenii (formerly B. subtilis subspecies spizizenii) and S. aureus as representative Gram-positive species. B. spizizenii is also a spore-forming bacterium. As can be seen from the data provided in Table 3 and Fig. 1, Chemgene HLD4L, Prevail™, and TriGene Advance were bactericidal against B. spizizenii at dilutions far exceeding the recommended dilution at all holding times. In contrast, with Effective Ratios less than one, bleach and Virkon required a lower dilution than that recommended to achieve bactericidal activity at all holding times (Fig. 1). For S. aureus, all decontamination agents were effective at or below the recommended dilution except for bleach with a holding time of 10 min. There was also a clear relationship between concentration and time, with all decontaminating agents being effective against S. aureus at higher dilutions as the holding time increased.

Activity of decontaminating agents against Mycobacteria

We tested four Mycobacterial species, M. abscessus, M. marinum, M. smegmatis, and M. tuberculosis, against the panel of decontamination agents (Table 3, Fig. 1). None of the agents were effective against M. abscessus at the recommended dilution, except for bleach at a holding time of 1,440 min. The data are similar for M. marinum, with the addition of Trigene Advance being effective at the recommended dilution for holding times of 60 min or more. M. smegmatis was the most sensitive of the four mycobacterial species tested, with both Chemgene HLD4L and Trigene Advance effective at or higher than the recommended dilution at all holding times. The remaining decontamination agents were also effective at or higher than the recommended dilution at holding times at 240 min and longer except for bleach which was also effective at 60 min. For M. tuberculosis, Trigene Advance was the decontamination agent effective at or higher than the recommended dilution at all holding times. Chemgene HLD4L was also effective above the recommended dilution but only with a holding time of 1,440 min.

Discussion

It is reassuring that many of the decontamination agents tested work above the parameters assessed in the standards against a wide range of bacterial species. Even if the decontamination agent works at a higher efficacy than is recommended by the manufacturer, it is still prudent to use the recommended dilution to ensure complete sterilisation. However, for some organism and decontamination agent combinations, we obtained an effective ratio below one indicating that the recommended dilution was not effective.

In many situations, the real-world usage of decontamination agents will differ from the conditions they were tested under using the standards. For example, neutralization agents are not used and the bacterial species being decontaminated may be growing in media containing substances capable of inhibiting the decontamination agents directly or mitigating their effects. In the worst-case scenario, this will result in a decontamination failure. For example, Prevail™ is advertised as an anti-mycobacterial agent with activity against M. tuberculosis with a contact time of just 5 min. Yet our data shows that it was only effective at the recommended dilution against the non-pathogenic M. smegmatis at holding times of 240 min or greater and was not effective against pathogenic species such as M. tuberculosis and M. abscessus. The enhanced resistance of Mycobacteria against many chemical agents is well known and has led to the development of separate standards for testing against Mycobacteria (Best et al., 1990; Griffiths, Babb & Fraise, 1998; Russell, 2001). Standards EN 14204 and EN 14348 evaluate the mycobactericidal activity of chemical disinfectants and antiseptics used in the veterinary area and in the medical area, respectively. One reason for the discrepancy between our results and the advertised activity of Prevail™ may be because M. smegmatis has less stringent nutritional requirements for growth so the testing conditions may not have recapitulated the standard growth conditions of other, arguably more important, species. Supplementation with catalase is common when growing many mycobacterial species including M. tuberculosis. As catalase mitigates against the toxic effects of hydrogen peroxide by converting it to water and oxygen, it is unsurprising this would have an impact on the activity of Prevail™ which is a hydrogen peroxide-based decontamination agent.

Our data serves as a warning that decontamination agents may not be effective against all strains of a particular species. In our study, we tested the efficacy of five decontamination agents against four strains of E. coli, including two antibiotic-sensitive and two antibiotic-resistant isolates. Biocides generally have non-specific targets which eventuate in cell death. As with other antimicrobials, resistance to decontamination agents can develop. Indeed, many organisms can develop natural resistance against certain compounds, such as how Pseudomonads have generally high intrinsic resistance against many agents including antibiotics (Adair, Geftic & Gelzer, 1969; Aires et al., 1999). Resistance or tolerance to a decontamination agent could have a natural genetic basis or be acquired through co-resistance with antibiotic resistance genes. Tattawasart and colleagues showed that some chlorhexidine-resistant strains of Pseudomonas stutzeri have cross-resistance against other biocides such as triclosan, as well as antibiotics including rifampicin and polymyxin B (Tattawasart et al., 1999). This cross-resistance between biocides and antibiotics was also identified with the use of triclosan and prolonged hydrogen peroxide treatment (Tattawasart et al., 1999; Wesgate, Grasha & Maillard, 2016).

Our data shows that the antibiotic-resistant E. coli strain BAA-2452 is more resistant to TriGene Advance than the antibiotic-sensitive quality control strain ATCC 25922, and more resistant to Virkon than the antibiotic-resistant strain BAA-2430. BAA-2340 and BAA-2452 are both carbapenem-resistant though they encode different carbapenemases. Resistance in BAA-2452 is attributed to the acquisition of the Klebsiella pneumoniae Carbapenemase (KPC) encoded by blaKPC. The KPC was initially acquired through transposon Tn4401 which encoded a β-lactamase that can hydrolyse carbapenems, a class of β-lactam based antibiotics resistant to degradation by other β-lactamases (Cuzon, Naas & Nordmann, 2011). In this case, it is unlikely that a β-lactamase would be involved in resistance against a quaternary ammonium decontamination agent. Bacterial membrane features and proteins are known to be involved in the intrinsic resistance of some organisms by creating a barrier to the agent or encoding an efflux mechanism to prevent the agent from reaching its target (Russell, 2001). The importance of this membrane barrier could be a reason why many of the commercial decontamination agents contain surfactants. This implies that the intrinsic properties of a bacterium may be more important than its antibiotic-resistance status. This is supported by our finding that bleach was less effective against the antibiotic-sensitive molecular cloning strain Top 10 at holding times less than 240 min. This strain was also more resistant to Chemgene HLD4L than ATCC 25922.

Another important consideration is the ability of some bacterial species and isolates to produce biofilms. These are typically characterised by microcolonies of bacterial cells encased in an extracellular polymeric substance (EPS) matrix (Flemming et al., 2016). While we did not explore this in our study, many groups have previously shown that bacteria growing in a biofilm are more resistant to a variety of decontaminating agents when compared to planktonically-grown cells (Bridier et al., 2011).

Conclusions

In conclusion, when deciding if a commercial decontamination agent is suitable for use, it would be sensible to consider how the standards used to test its efficacy relate to the application that the decontaminant is being considered for. As the responsibility for correct usage is with the end user, informed decisions regarding the choice of decontamination agent can only be made if those details are readily available, such as on the concentrate bottle. However, these details are often not available, or the recommendations on the label are different to how the product was tested. It is therefore imperative that manufacturers make the conditions of the testing available to product users. Here we show that the Minimum Bactericidal Dilution Factor (MDBF) assay and controlled neutralization by dilution can be used to determine the activity of a blend of compounds with multiple active ingredients. Considering our findings that species within the same genus and strains within a species can differ in their susceptibility to a variety of decontamination agents, we advise that users verify the efficacy of decontamination agents under the conditions in which they will be used to ensure that laboratory materials will be properly decontaminated. Our data shows that merely increasing the recommended hold time rather than carrying out a proper validation is not sufficient for all organism-decontamination agent combinations.

The authors would like to thank David Jenkins for his encouragement and support to carry out this study.

Additional Information and Declarations

Competing Interests

Author Contributions

Data Availability

Siouxsie Wiles is an Academic Editor for PeerJ.

Benedict Uy conceived and designed the experiments, performed the experiments, analyzed the data, prepared figures and/or tables, authored or reviewed drafts of the article, and approved the final draft.

Hannah Read conceived and designed the experiments, performed the experiments, analyzed the data, authored or reviewed drafts of the article, and approved the final draft.

Shara van de Pas analyzed the data, prepared figures and/or tables, and approved the final draft.

Rebecca Marnane analyzed the data, authored or reviewed drafts of the article, and approved the final draft.

Francesca Casu analyzed the data, authored or reviewed drafts of the article, and approved the final draft.

Simon Swift analyzed the data, authored or reviewed drafts of the article, and approved the final draft.

Siouxsie Wiles analyzed the data, prepared figures and/or tables, authored or reviewed drafts of the article, and approved the final draft.

The following information was supplied regarding data availability:

The data is available at Figshare: Wiles, Siouxsie; Uy, Benedict; Read, Hannah; Swift, Simon; Marnane, Rebecca; Casu, Francesca (2022): The efficacy of commercial decontamination agents against a variety of bacterial species in a research laboratory setting. The University of Auckland. Dataset. https://doi.org/10.17608/k6.auckland.19142606.v1.

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
