# Peer review of "The efficacy of commercial decontamination agents differs between standardised test settings and research laboratory usage for a variety of bacterial species"

_PeerJ, doi:10.7717/peerj.13646_

## Round 0.1 · original submission · Major Revisions

We have received two detailed review reports. Reviewer #1 is positive but reviewer#2 has very serious concerns regarding your methodology, and addressing them will require a complete repetition of your work using a proper neutralization step after the incubation with the biocidal agents. I agree that, without that step, one cannot be sure whether one is really observing biocidal rather than bacteriostatic effects. I am sorry I cannot have better news for you, but I expect both reports to be very helpful towards making your work as reliable as you intend.

Reviewer 1 ·

Basic reporting

line 42: change active to effective. this applies throughout paper.
44: Suggest referring to disinfectants by their active ingredient rather than brand names, this applies throughout the article.
141: define BD BBL. Make sure all abbreviations and acronyms are defined throughout text.
154: add reference for the MBDF assay
220: need to describe how AUC was determined in the Methods section
247 and elsewhere: the word "data" is plural, so change the word "is" that follows to "are". this applies throughout
287: add reference for selection of antibiotic resistant E coli
428-435 and 438-446: are these paragraphs supposed to be in figure legend? If so they are way too long and should be in narrative for Results or Discussion
Table 2: normally the manufacturer's location for where it is purchased is included

Experimental design

130: indicate source of bacteria, i.e., where were they purchased
143: with such a wide range in bacteria levels, how can you be sure that differences in results you saw were not attributed to differences in starting bacterial levels. it would've been better to try to maintain similar levels regardless of species.
156: so duplicates were used for each bug/dilution/disinfectant? Need to clarify number of replicates used.
164: indicate what the appropriate agar was
general comment: consider adding some text to explain how the assay you used is different than some of these standards that you mention

Validity of the findings

38: these really aren't real world conditions. these are just simple in vitro tests. Tests would've been much more realistic if actual materials were used instead of using suspension tests.
214: indicate whether the minor differences were statistically significant
215: indicate the reference or source indicating which dilution should be used for a generic chemical such as bleach

Additional comments

Figure 2 legend: indicate what the units are for AUC, and how calculated. And for the whisker plots, need to define the statistics associated with the boxes and whiskers.

Reviewer 2 ·

Basic reporting

Writing is generally of good quality and is written understandably. The cited literature is appopriate and provides a good background to the work and the raw data is available online.

The figures are presented professionally although I found the results sometimes to be a bit difficult to interpret (particualrly area under curve data). However, I am unfamilar with this type of analysis so this may just be me.

Experimental design

I think the aims could be explained and justified more clearly. The authors mention that large/heat sensitive items cannot be autoclaved but the study itself seems to only be concerned on the efficacy of various disinfectants for decontaminating purified laboratory cultures (which can easily be autoclaved).

The experimental design has a major flaw in that the authors did not seem to utilise a neutralisation agent (e.g., Dey-Engley/SCDLP broth) to quench the activity of the biocides following completion of the contact time. All EN standards require the biocides to be neutralised using an agent which is validated for both efficacy and toxicity. Without a neutralisation step, contact times may actually be considerably longer than what was stated. Additionally, the biocide would still be present when the bacteria were enumerated; as biocidal concentrations were being used, this would almost certainly mean that some of the biocides would have remained above the minimum inhibitory concentration. This means that observing "no growth" could simply be due to growth inhibition, rather than bactericidal activity.

Biocides were also diluted in Mueller Hinton or 7H9 broth prior to use. These broths contain proteins which would seriously interfere with the efficacy of the formulations.

One of the authors' critiques of current testing standards is that products only have to demonstrate a 5 log10 reduction to be certified as effective. However, by my calculations (provided in the attached document), their assay is only able to determine a 4.15 log10 reduction for mycobacteria, since the lower limit of detection of their enumeration method was not taken into account.

Validity of the findings

Unfortunately the lack of neutralisation completely invalidates the findings of the study, as none of the data can be trusted to properly reflect the reported test conditions. I have provided a more detailed breakdown in the attached feedback.

Annotated reviews are not available for download in order to protect the identity of reviewers who chose to remain anonymous.

---

## Round 0.2 · accepted · Accept

I have read your response and the changes introduced in the manuscript. I agree that you have addressed our reviewers' criticism and I am glad to accept your paper for publication. I apologize for the time taken to reach my decision, which was due to the need to wait for the reinvited reviewers to signal their (un)availability to perform a new review.